# Utilization of Modified Sunflower Seed as Novel Adsorbent for Nitrates Removal from Wastewater

Antonija Kristek Janković [1], Mirna Habuda-Stanić [2,*], Huiyu Dong [3], Ana Tutić [4], Željka Romić [5], Maja Ergović Ravančić [6], Tibela Landeka Dragičević [7] and Mario Šiljeg [8]

[1] Tourism and Catering School Osijek, Ul. Matije Gupca 61, 31000 Osijek, Croatia; antonija.kristek-jankovic@skole.hr
[2] Faculty of Food Technology Osijek, Josip Juraj Strossmayer University of Osijek, Franje Kuhača 18, 31000 Osijek, Croatia
[3] Key Laboratory of Drinking Water Science and Technology, Research Center for Eco-Environmental Sciences, Chinese Academy of Sciences, Beijing 100085, China; hydong@rcees.ac.cn
[4] Bor-Plastika d.o.o. Glavna 2, 31309 Kneževi Vinogradi, Croatia; ana.tutic53@gmail.com
[5] Vodovod-Osijek d.o.o. Poljski put 1, 31000 Osijek, Croatia; zeljka.romic@vodovod.com
[6] Faculty of Tourism and Rural Development in Požega, Josip Juraj Strossmayer University of Osijek, Vukovarska 17, 34000 Požega, Croatia; mergovic@ftrr.hr
[7] Faculty of Food Technology and Biotechnology, University of Zagreb, Pierottijeva 6, 10000 Zagreb, Croatia
[8] Josip Juraj Strossmayer Water Institute, Ulica grada Vukovara 220, 10000 Zagreb, Croatia; siljeg.mario@gmail.com
* Correspondence: mirna.habuda-stanic@ptfos.hr; Tel.: +385-31-224-326

**Abstract:** The food processing waste, sunflower seed shells (SS), were chemically modified and tested as adsorbents for nitrate removal from water and wastewater. Chemical modification of the SS implied a quaternization reaction. Efficiency and mechanisms of nitrate removal from water by modified sunflower seed shells (MSS) were examined using model nitrate solution (MS) and samples of real wastewater (RW) in batch adsorption experiments while the regeneration capacity was tested by fixed bed adsorption column and regeneration experiments. The MSS had the highest nitrate adsorption capacity of 12.98 mg g$^{-1}$ for model nitrate solution, 12.16 mg g$^{-1}$ for model wastewater, 13.70 mg g$^{-1}$ for the wastewater generated by the confectionery industry (CI), and 12.52 mg g$^{-1}$ for the wastewater generated from the meat industry (MI). Equilibrium data were analyzed, and results demonstrated a better fit for the Freundlich isotherm model, while kinetic models showed that the adsorption has pseudo-second-order kinetics. Adsorption and desorption experiments in fixed bed columns showed good MSS regeneration performances and efficiency after a minimum of three cycles. Lower efficiencies of regenerated MSS were noted when real confectionery and meat industry effluent were treated. Environmental toxicity of nitrate saturated MSS was tested using an acute toxicity test with freshwater plankton *Daphnia magna*. SS showed very good properties and could be competitive among already known and existing "low-cost" adsorbents as potential adsorbents for nitrate removal from water and wastewater.

**Keywords:** nitrate removal; wastewater; adsorption; sunflower seed shells; chemical modification

## 1. Introduction

Water resources in many countries are very often polluted with nitrogen compounds that cause environmental disturbances such as eutrophication and biodiversity loss. In addition, it can cause several health problems such as vomiting, hypertension, diarrhea, and "blue baby" syndrome. Nitrogen compounds enter water resources through municipal wastewater but also through industrial wastewater, where significant amounts of nitrogen compounds are often found in wastewater from the food industry [1]. Since the nitrate ion is extremely dissoluble in water and is unable to connect normally with soil particles, it can easily enter subterranean and surface water [2]. Extortionate use of water with high

nitrate concentrations can induce health issues like hypertension, diabetes, breathing and stomach diseases, and methemoglobinemia [3]. Concerning the concentration of nitrate in drinking water, the World Health Organization set the maximum concentration level at 50 mg L$^{-1}$ [4]. Hence, constant upgrading of the nitrate displacement technologies from table water, as well as from wastewater before its release into the natural environment, is of great significance [5].

Before discharge into the environment, wastewater can be treated by applying several physicals, biological, and chemical procedures; the use of each procedure is conditioned by the initial quality of the wastewater. Various techniques have been used for nitrate removal from water and wastewater, among which membrane filtration, electrodialysis, ion exchange, biological denitrification, and adsorption showed efficient results [2].

Adsorption is recognized as one of the most valuable methods [3]. Nowadays, scientists are focused on the green economy and green technologies. In that context, many agricultural and industrial byproducts are tested as adsorbents in water treatment processes due to their renewability, low cost, and accessibility as part of the solution in waste management and reducing negative environmental impact. Additionally, these materials could be transmuted into highly valuable adsorbents, creating a simple source and highly successful substance for applications in the environment [6]. Of all the previously mentioned procedures, adsorption was recognized as the most appropriate procedure usually utilized at an industrial scale; this is because of its operation and simple design, its adjustable nature, and because it is effective, senseless to toxic substances, and renewable [7,8].

Despite these benefits, adsorption also has some disadvantages, such as the problem of saturated adsorbent disposal and extended conventional detachment processes for the separation of adsorbent from water solution [9]. Nitrate removal from water solutions by adsorption has been proposed as an efficient procedure [3]. Therefore, the development of low-cost but effective adsorbents from different waste substances could be a promising alternative. Various types of natural waste and materials have been utilized to adsorb nitrates from water, including sugarcane bagasse, zeolite, fly ash, clays, rice husk, chitosan, and slag, but no adsorbent has shown amazing performance for higher levels of adsorption capacity [2,6]. The adsorption capacities of several low-cost adsorbents towards nitrate are presented in Table 1.

**Table 1.** Adsorption capacities of some low-cost adsorbents towards nitrate.

| Adsorbent | Nitrate Adsorption Capacity | Reference |
|---|---|---|
| Modified hazelnut shells | 25.79 mg g$^{-1}$ | [10] |
| Franco biochar | 1.339 mg g$^{-1}$ | [11] |
| Modified brewers' spent grain | 22.65 mg g$^{-1}$ | [12] |
| Modified reed straw | 272.024 mg g$^{-1}$ | [13] |
| Modified olive mill residues | 110 mg g$^{-1}$ | [14] |
| Modified grape seeds | 25.626 mg g$^{-1}$ | [15] |
| Modified corn stalks | 23.59 mg g$^{-1}$ | [16] |
| Modified bambo chopstick | 16.39 mg g$^{-1}$ | [17] |
| Modified lychee peels | 60.3 mg g$^{-1}$ | [18] |
| Modified corn-cob | 9.35 mg g$^{-1}$ | [19] |
| Modified rice husk ash | 30.86 mg g$^{-1}$ | [20] |

During the past decade, the rising interest in biomass application has encouraged several studies and prompted examinations of lignocellulosic materials as potential adsorbents in water treatments. Lignocellulosic materials, such as pomace, fruit peels, sawdust, straw, and tree bark, were found to be efficient adsorbents after chemical modification as cationization [2,21,22]. Only a few modification methods, which include the integration of quaternary ammonium groups, have been presented for this purpose. Using epichlorohydrin and dimethylamine in the presence of pyridine, Orlando et al. [23] first reported successfully synthesized ion exchange resins from agricultural waste (rice hull) with a higher adsorption capacity of 16.9 mg g$^{-1}$ for NO$_3$-N [24].

Ighalo et al. [25] recently investigated the costs of biomass-based adsorbent preparation as one of the key factors of its application in water treatment processes. Using a design metric model, the authors expressed the adsorption capacities of biomass-based adsorbents in USD/mol. Based on this metric model, authors quoted that most adsorbents' cost performance is between 1 and 200 USD/mol. They grouped adsorbents as "acceptable for usage" if their usage price is <1 USD/mol, while adsorbents with a usage price >200 USD/mol are marked as expensive and their usage as unreasonable.

The waste lignocellulosic material used in this study, sunflower seed shells (SS), provides the standard for cheap adsorbents. SS is a byproduct of the oil industry, which produces up 30% of sunflower husks. During oil production, sunflower seed shells are considered worthless products and, therefore, burned as a source of heat. In our case, the local oil factory uses sunflower seed shells for boiler heating; the remainder is available for the market with an average price of 50 USD/ton.

Sunflower husk is rich in cellulose and hemicellulose, and they are fundamental for the processes of quaternization. Sunflower husk was also used as an adsorbent for removing cationic dyes and various heavy metals and for removing copper ions from wastewater. The obtained results introduced sunflower seed husk as a cost-effective and efficient adsorbent for copper removal from wastewater [26].

This paper presents the results of a study conducted to test nitrate removal from the aquatic environment using SS modified with the process of quaternization. In experiments with batch adsorption, the effect of key process parameters on nitrate adsorption by MSS were tested (initial $NO_3$-N concentration, pH, contact time, and adsorbent concentration). In addition, the efficiency of nitrate removal by MSS was validated using a fixed bed column and regeneration test. The environmental impact of discharged nitrate-saturated MSS was tested using an acute toxicity test with *Daphnia magna*.

## 2. Materials and Methods

### 2.1. Materials

All used chemicals were analytical grade. N,N-dimethylformamide was obtained from GramMol (GramMol, Zagreb, Croatia). Ethylenediamine and epichlorohydrin were obtained from Sigma Aldrich (Sigma Aldrich, St. Louis, MO, USA), and triethylamine from Fisher Scientific (Leicestershire, UK). For the preparation of nitrate anion, $KNO_3$ (Merck, Darmstadt, Germany) was used. The initial stock solution of 1000 mg $L^{-1}$ (as N-$NO_3^{-}$) was prepared as follows: 7.218 g $KNO_3$ was dissolved in 1 L of demineralized water.

Tested initial concentrations for the experimental solutions ranging from 10 to 300 mg $L^{-1}$ were made by watering down the stock solution. Model wastewater was made due to the procedure described by Kosjek et al. [24] and a proper amount (10–30 mg $L^{-1}$) of $KNO_3$ solution. From the local confectionery and meat industry, 24-h mixture samples of real wastewater were collected. The physical and chemical characteristics of the model and real wastewater obtained from the confectionery and meat industries are presented in Table 2. For the determination of the physicochemical properties of model wastewater and real wastewater, Standard Methods were used (ISO 6060:1989 [27], ISO 5663:2001 [28], ISO 7890-3:1988 [29], HRN EN 26777:1998 [30], ISO 6878:2004 [31], and ISO 10523:1998 [32]). All chemicals used in this study were of analytical grade.

**Table 2.** Characteristics of used model wastewater and wastewaters from the confectionery and meat industry.

| Parameters | Model Wastewater * | Confectionery Industry Wastewater | Meat Industry Wastewater |
|---|---|---|---|
| COD (mg$O_2$ $L^{-1}$) | 785 | 14,488 | 1200 |
| $N_{total}$ (mg $L^{-1}$) | 330 | 83 | 48 |
| N-$NH_4$ (mg $L^{-1}$) | 25 | 35 | 8 |
| N-$NO_3$ (mg $L^{-1}$) | 2.45 | 50 | 65 |
| N-$NO_2$ (mg $L^{-1}$) | <0.002 | <0.002 | 0.45 |

**Table 2.** *Cont.*

| Parameters | Model Wastewater * | Confectionery Industry Wastewater | Meat Industry Wastewater |
|---|---|---|---|
| P-PO$_4$ (mg L$^{-1}$) | 27.51 | 16 | 42 |
| pH | 7.48 | 5.7 | 9.4 |
| Colour | yellowish | yellow-brown | gray-brown |

* Not spiked with NO$_3^-$.

## 2.2. Adsorbent Preparation

The sunflower seed shells were obtained from Oil Factory Čepin Ltd. (Čepin Croatia). The samples of SS were milled with a laboratory knife mill using a 1 mm screen (MF10 basic, IKA Labortechnik, Staufen im Breisgau, Germany) and then processed using a vibratory sieve shaker (AS 200 Digit, Retsch GmbH, Haan, Germany). Obtained samples were analyzed and results showed that the dominant sieved particle fraction is in the range of 200 to 315 μm. Therefore, this fraction was used for further study.

Chemical modification of SS was carried out using epichlorohydrin-triethylamine method, also known as the ETM method, described in detail by Keränen et al. [33]. Initially, 2 g of SS was mixed with 16 mL of N,N-dimethylformamide (DMF) and 13 mL of epichlorohydrin (ECH) for 45 min at a temperature of 70 °C. The addition of ethylenediamine (2.5 mL) followed and the mixture was further stirred at 80 °C for 45 min. Then, the formation of amine groups was caused by the addition of 13 mL of triethylamine. Stirring continued for a 120 min at 80 °C. The obtained MSS were then washed with demineralized water and dried at 100 °C for 24 h. The above-described preparation route is shown in Scheme 1.

**Scheme 1.** Preparation route of modified SS.

## 2.3. Structural Characterization of Raw and Modified Sunflower Seed Shells

In this study, five different materials were structurally characterized: (a) raw sunflower seed shells (SS), (b) modified sunflower seed shells (MSS), (c) modified sunflower seed shells after nitrate adsorption from nitrate model solution (MSS-N), (d) modified sunflower seed shells after nitrate adsorption from confectionary wastewater (MSS-Nc), and (e) modified sunflower seed shells after nitrate adsorption from meat industry wastewater (MSS-Nm).

Structural analyses were conducted to determine the elemental composition of all five materials. The content of the following elements was determined: C, O, N, and Cl by energy-dispersive X-ray spectroscopy (EDS) using X-ray photoelectron spectroscopy (XPS, EscaLab 250Xi). The surface structures and morphology, effects of modification, and nitrate adsorption on MSS were examined using Field Emission Scanning Electron Microscope (SEM) with a semi-in-lens and Gentle Beam (SEM, JEOL, JSM-7001 F). A high-resolution

tool Transmission Electron Microscope (TEM) was used to analyze atomic arrangement in the above mentioned five materials (TEM, Hitachi, H-7500).

*2.4. Batch Adsorption Experiments*

Batch adsorption experiments were performed to define the efficiencies of nitrate adsorption from water samples onto MSS regarding the following parameters: MSS concentration, initial N-NO$_3$ concentration, initial pH value, contact time, and temperature.

The effects of MSS mass on nitrate removal were examined using various quantities of MSS, ranging from 0.05 g to 0.5 g. The experiments were conducted using 50 mL of solution with an initial N-NO$_3$ concentration of 30 mg L$^{-1}$ at 25 °C for 120 min.

The effect of the initial nitrate concentration on the adsorption efficiency process was tested using the solution with an N-NO$_3$ concentration range from 10 to 200 mg L$^{-1}$. Tests were performed at 25 °C for 120 min using 0.4 g L$^{-1}$ of the MSS.

The effects of initial pH on nitrate removal by MSS were analyzed within a pH range from 2.0 to 10.0. The tests were performed using the solution with an initial N-NO$_3$ concentration of 30 mg L$^{-1}$ and an adsorbent concentration of 4 g L$^{-1}$ at 25 °C for 120 min.

To examine the efficiency of nitrate removal regarding the contact time, adsorption tests were performed using an adsorbent mass of 4 g L$^{-1}$ at 25 °C. Tests were performed under initial pH values of model N-NO$_3$ solution, MW and RW from the confectionery and the meat industry, i.e., at 6.55, 5.77, 5.3, and 6.81, respectively.

The effects of temperature on nitrate removal were tested at 25, 35, and 45 °C. All tests were performed in a thermostatic shaker (Bioblock Scientific, Polytest 20, Illkirch, France) at 130 rpm. For pH adjustment, 0.1 mol L$^{-1}$ NaOH and 0.1 mol L$^{-1}$ HCl solutions were used. After adsorption, all samples were filtered using 0.45 pore size filters. The residual nitrate concentrations were determined prior and after adsorption experiments by UV/Vis spectrophotometer (Specord 200, Analytic Jena, Jena, Germany) at 410 nm using the ISO 7890-3:1998 method [18]. All tests were performed in duplicate and found to be repeatable. Obtained data were processed and analyzed using the software system Statistica 13.3 (Statsoft Inc., St Tulsa, OK, USA).

After each test, the percentage of nitrate removal $R$ (%) was calculated using the equation:

$$R = ((\gamma_0 - \gamma)/\gamma_0) \cdot 100 \tag{1}$$

where $\gamma_0$ (mg L$^{-1}$) is the initial nitrate concentration and $\gamma$ (mg L$^{-1}$) is the final nitrate concentration after the adsorption test.

The quantity of adsorbed nitrate by MSS at the equilibrium was calculated by the equation:

$$q_e = \frac{(\gamma_0 - Y_e) \cdot V}{m} \tag{2}$$

where $q_e$ was the quantity of nitrate adsorbed by MSS (mg g$^{-1}$), $\gamma_0$ was the initial nitrate concentration (mg L$^{-1}$), $\gamma_e$ was the residual nitrate concentration (mg L$^{-1}$), V was the solution volume (L), and m was the used mass of MSS (g).

The results of batch adsorption experiments were investigated using Langmuir and Freundlich adsorption models. Several kinetic models, like pseudo-first-order, pseudo-second-order, and intraparticle diffusion models were utilized to analyze the kinetics of nitrate adsorption by MSS.

*2.5. Fixed-Bed Column Regeneration Test*

The fixed-bed column regeneration tests were carried out utilizing a glassy tube of 13 mm internal diameter and 220 mm of total length loaded with 1 g MSS. From the top of the column, 2000 mL of nitrate solution (30 mg L$^{-1}$) was fed at a controlled constant flow rate of 10 mL min$^{-1}$ using a peristaltic pump (Masterfelx L/S, Cole-Palmer Instrument Company, Vernon Hills, IL, USA). The tests were performed at room temperature with no prior pH adjustment. All regeneration tests were started by the leak of the demineralized water through the column and immersion of the MSS and trapped air displacement. Then,

250 mL of effluent was collected at the bottom of the column and the nitrate concentration was determined using the method described in Section 2.4.

Adsorbent regeneration was carried out using 200 mL of 0.1 M NaCl followed by 500 mL of demineralized water at a flow rate of 10 mL min$^{-1}$. The saturation capacity $q_s$ (mg g$^{-1}$) was determined using the equation:

$$q_s = \frac{\gamma_0 V_0 - \sum \gamma_n V_n}{m} \tag{3}$$

where $\gamma_0$ presented the initial nitrate concentration (mg L$^{-1}$), $V_0$ was the volume of used nitrate solution (L), $\gamma_n$ was the final nitrate concentration in fraction $n$ (mg L$^{-1}$), $V_n$ presented the volume of fraction $n$ (L), and $m$ is the quantity of MSS (g).

### 2.6. Adsorption Equilibrium Modeling

Results obtained by batch adsorption tests were analyzed using the two most used adsorption isotherm models, the Langmuir and Freundlich isotherm models, that explain the distribution of adsorbed compounds between liquid and solid phases at equilibrium. The adsorption isotherm models provide data about the adsorption mechanism, and this is important for the practical use of adsorbents [33,34].

### 2.6.1. Langmuir Isotherm Model

The Langmuir isotherm is derived on assumption that (i) the adsorption area is limited, (ii) the adsorbent is structurally homogeneous, (iii) at equilibrium, the number of molecules being adsorbed will be equal to the number of molecules leaving the adsorbed state, and (iv) a monolayer adsorption process. Therefore, the well fit of obtained data with the Langmuir model implies monolayer adsorption. The linear form of the Langmuir equation is:

$$\frac{\gamma_e}{q_e} = \frac{q_m K_L \gamma_e}{1 + K_L \gamma_e} \tag{4}$$

where $q_e$ presents the quantity of adsorbed nitrate per certain quantity of adsorbent (mg g$^{-1}$), $\gamma_e$ is the equilibrium concentration (mg L$^{-1}$), $q_m$ is the maximum quantity of nitrate ions needed to form a monolayer (mg g$^{-1}$), and $K_L$ (L mg$^{-1}$) is the Langmuir constant. The values of $q_m$ and $K_L$ can be defined from the linear image of $\gamma_e/q_e$ vs. $\gamma_e$ [35]. Additionally, the separation factor ($R_L$) as a dimensionless constant can be expressed as:

$$R_L = \frac{1}{1 + K_L \gamma_0} \tag{5}$$

where $\gamma_0$ is the initial adsorbate concentration (mg L$^{-1}$). The Langmuir isotherm model can be rated as suitable if $0 < R_L < 1$ [36].

### 2.6.2. Freundlich Isotherm Model

The Freundlich isotherm model of adsorption is derived from assumption that (i) the adsorbent contains a heterogeneous surface, (ii) the adsorption is reversible, and (iii) the adsorption can be in the monolayer and/or multilayer formation [34,37,38]. The linear equation of the Freundlich isotherm model is:

$$\ln q_e = \ln K_F + \frac{1}{n} \ln \gamma_e \tag{6}$$

where $K_F$ is the Freundlich constant ((mg g$^{-1}$)(mg L$^{-1}$)$^{-1/n}$) and its larger value implies a higher capacity of adsorbent, while the constant $1/n$ is a function of the adsorption strength. A smaller value of $1/n$ implies a strong adsorption bond, i.e., adsorption intensity. Equilibrium constants can be estimated from the linear image of $\ln q_e$ vs. $\ln \gamma_e$.

*2.7. Adsorption Kinetic Modeling and Mechanism*

The adsorption kinetics modeling is an efficient tool for the prediction of the reaction pathway and mechanism of the process of adsorption [39]. The kinetic data of the nitrate adsorption by MSS was analyzed by:: pseudo-first order, pseudo-second order, energy activation, and intraparticle diffusion model, using the following equations:

Pseudo-first order model:

$$\log(q_e - q_t) = \frac{\log q_e - k_1}{2.303} \tag{7}$$

Pseudo-second-order model:

$$\frac{t}{q_t} = \frac{1}{k_2 q_e^2} + \frac{t}{q_e} \tag{8}$$

Energy activation:

$$lnk_2 = lnA - \frac{E_a}{RT} \tag{9}$$

Intraparticle diffusion model:

$$q_t = k_i t^{1/2} + C \tag{10}$$

where $q_e$ and $q_t$ (mg g$^{-1}$) present the nitrate content adsorbed at equilibrium, respectively, and time $t$. The kinetic rate constants for the pseudo-first and pseudo-second-order models are presented as $k_1$ (min$^{-1}$) and $k_2$ (g mg$^{-1}$ min$^{-1}$), respectively, while the intraparticle diffusion rate constant is $ki$ (mg g$^{-1}$ min$^{-0.5}$). The pre-exponential factor is presented by $A$ while Ea presents energy activation and $R$ stands for the universal gas constant of 8.314 J mol$^{-1}$ K$^{-1}$. The value of parameter $C$ (mg g$^{-1}$) provides information about the boundary layer and the higher value of $C$ implies a stronger boundary layer effect on the adsorption mechanism [3].

All equations used data obtained from adsorption tests conducted using $\gamma_{\text{adsorbent}}$ = 4 g L$^{-1}$ of MSS, an initial nitrate concentration of $\gamma_{\text{nitrate}}$= 30 mg L$^{-1}$, and a natural solution pH at 25, 35, and 45 °C.

*2.8. Determination of Acute Toxicity Using Daphnia Magna*

Ecotoxicological testing was conducted using nitrate-saturated MSS after adsorption tests conducted by (i) model nitrate solution, (ii) meat industry effluent wastewater, and (iii) confectionery industry effluent wastewater in order to determine acute toxicity i.e., determine the inhibition of the mobility of freshwater plankton *Daphnia magna*. The test includes the determination of plankton immobilization after exposure to the sample for 24 or 48 h under conditions specified by the standard method ISO 6341:2012 [40].

A weight of 2 g of an MSS used for adsorption of nitrates from model nitrate solution, meat industry effluent wastewater, and confectionery industry effluent wastewater was added to Erlenmeyer flasks and mixed with 100 cm$^3$ of demineralized water. The flasks were then placed on a shaker (mixing speed v = 150 rpm$^{-1}$) and stirred for 24 h. The sample was then filtered and centrifuged for 10 min at 3500 rpm$^{-1}$. The filtrate obtained in this way was used to determine the acute toxicity to freshwater plankton *Daphnia magna*. Dilutions of 1% and 0.5% solutions were prepared with demineralized water.

**3. Results and Discussion**

*3.1. Structural Characterization of Raw and Modified Sunflower Seed Shells*

The removal of nitrate was tested using a novel adsorbent prepared using sunflower seed shells and ETM method described in Section 2.2.

The content of C, H, N, and O were determined by energy-dispersive X-ray spectroscopy. In Table 3, the results of elemental analyses of the unmodified raw sunflower seed shells and MSS are presented.

**Table 3.** Elemental contents of raw (SS) and modified sunflower seed shells (MSS).

| Element | Atomic No. | SS | | | MSS | | |
|---|---|---|---|---|---|---|---|
| | | Weight (%) | Atomic (%) | Abs. Error (1 sigma) | Weight (%) | Atomic (%) | Abs. Error (1 Sigma) |
| C | 6 | 68.55 | 76.22 | 7.49 | 32.97 | 67.52 | 5.59 |
| N | 7 | 4.27 | 4.07 | 0.86 | 4.16 | 7.30 | 1.52 |
| O | 8 | 22.80 | 19.03 | 2.86 | 5.38 | 8.28 | 1.31 |
| Cl | 17 | 0.04 | 0.01 | 0.01 | 19.38 | 13.45 | 0.69 |
| Cu | 29 | 0.59 | 0.12 | 0.08 | 0.69 | 0.27 | 0.14 |

As was expected, caused by the chemical modification and ammonium group introduction during the modification method, the nitrogen content of MSS (7.30%) was much higher than the nitrogen content of the unmodified SS (4.07%). A similar result was reported by Stjepanović et al. [35]. Authors modified brewers' spent grain by quaternization and reported increases in nitrogen content from 3.6% for raw material to 9.42% for modified brewers' spent grain. Xu et al. [39] also reported an increase in nitrogen content as the result of the same modification process of lignocellulose materials. They reported that modification of the cotton stalk enlarged the nitrogen content from 0.32 to 3.74%, while, with modification of the wheat stalk, nitrogen content increased from 0.35 to 4.34%. A high increase of nitrogen content in modified material reported was reported by Keränen et al. [33]. Using the ETM modification, they reported an increase in the nitrogen content for spruce bark from 1 to 9%, for pine bark from 1 to 9.1%, for pine sawdust from 0.8 to 9.4%, birch bark from 0.9 to 9.2%, and peat from 1.9 to 9.8%.

FESEM micrographs are presented in Figure 1. From Figure 1, it can be observed that the surface of SS (Figure 1a) is more homogenous when compared to MSS (Figure 1b), on which cavities, micropores, and the rugged surface formed as the result of the ETM modification.

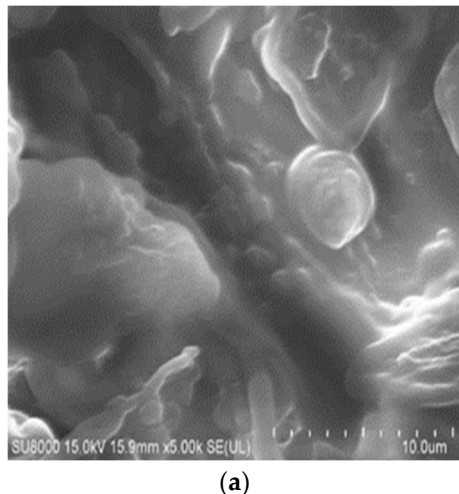

(**a**)

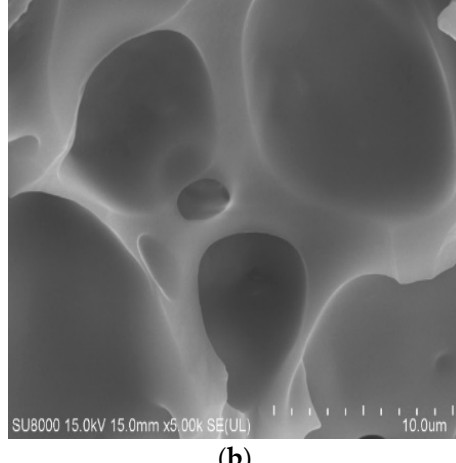

(**b**)

**Figure 1.** *Cont.*

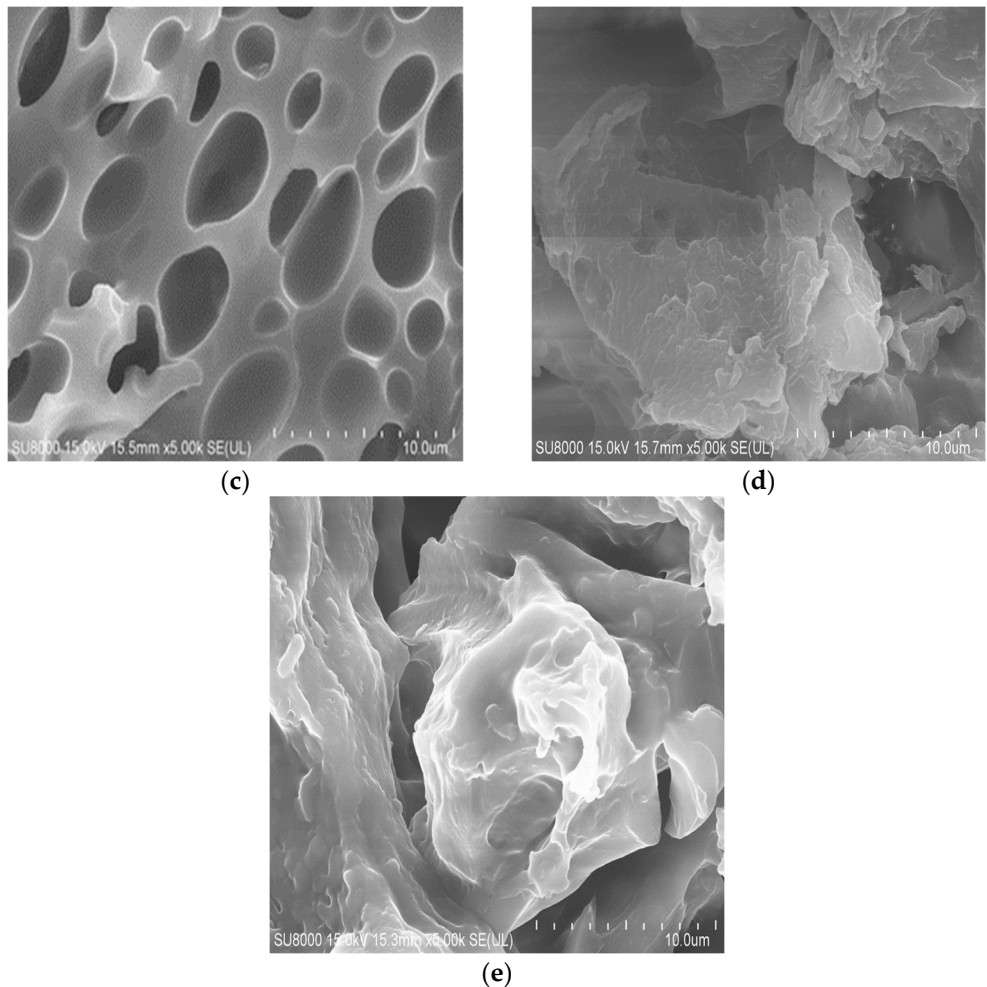

**Figure 1.** FESEM micrographs (magnification 5000) from (**a**) SS, (**b**) MSS, (**c**) MSS-N, (**d**) MSS-Nc, and (**e**) MSS-Nm.

Figure 1c,d,e show surfaces of MSS used for nitrate uptake. The similarity of the surface structure can be observed between the MMS used for nitrate removal from real wastewater samples (Figure 1d,e) since they contained other contaminants except for nitrates (Table 2), while Figure 1c shows only nitrate attached to the modified surface of MSS.

### 3.2. Batch Adsorption Experiments

Batch adsorption experiments were conducted to define the adsorption capacity of MMS under various MSS concentrations, initial $N\text{-}NO_3$ concentration, initial pH value, contact time, and temperature. Tests were conducted using (I) nitrate model solution, (II) model wastewater, (III) confectionary wastewater, and (IV) meat industry wastewater.

### 3.2.1. Effect of MSS Concentration on Nitrate Removal

The adsorbent dose is one of the most important adsorption factors that determines the amount of pollutant removal from water. Batch adsorption experiments can define the optimal amount of adsorbent required to remove a pollutant from water solution, which depends on the surface and number of active places accessible for adsorption. Information about optimal adsorbent dosage is important for optimizing and scaling-up the water treatment process. Figure 2 shows the graph of the effects of MSS concentration on nitrate removal. The tests were performed using adsorbent concentrations ranging from 1 to 10 g $L^{-1}$. From the presented results, it can be observed that the percentage of nitrate removal increases with the

adsorbent amount increase. The most efficient nitrate removal (92.36 %) was obtained when a combination of model nitrate solution and the highest adsorbent concentration of 10 g L$^{-1}$ was tested, while 80.77% of nitrate ions were removed in an experimental set that used MW.

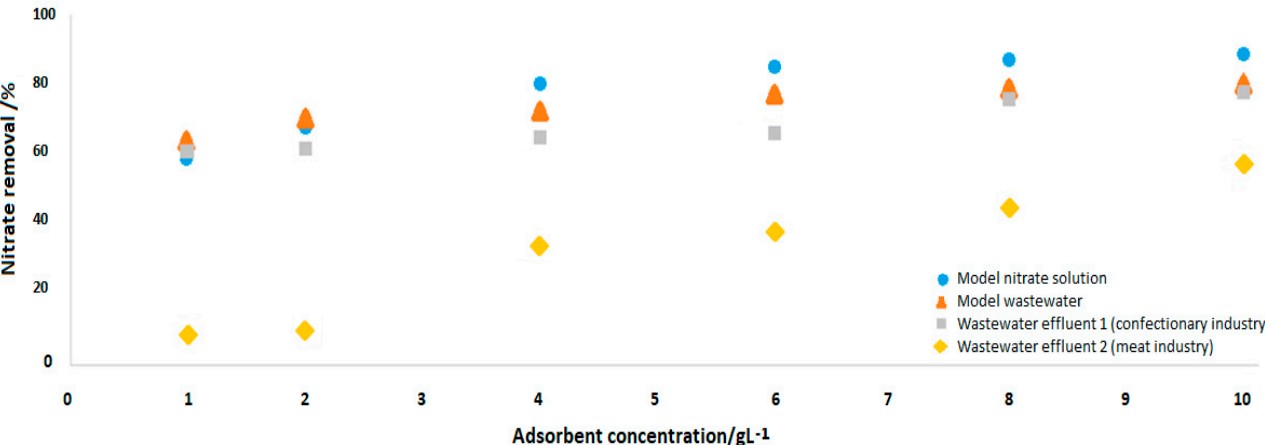

**Figure 2.** Effects of MSS concentration on nitrate removal (Y$_{nitrate}$ = 30 mg L$^{-1}$, T = 25 °C, V$_{nitrate\ solution}$ = 50 mL, t = 120 min).

In the sets of nitrate adsorption tests with MSS and real wastewater samples, the highest nitrate removal (78.24%) was obtained when confectionary industry wastewater was treated with 10 g L$^{-1}$ of MSS, while maximum nitrate removal (57.29%) was obtained in the treatment of meat industry wastewater when the same adsorbent weight and experimental conditions. From Figure 2, it can be also noted that a significant increase of adsorbed nitrate ions occurred up to MSS concentrations of 4 g L$^{-1}$, especially when the model nitrate solutions were treated, and a further increase of MSS concentration was not linearly followed by the rate of nitrate removal.

Many adsorption studies reported similar results; this confirms this phenomenon of a significant increase of nitrate and other pollutants' adsorption rate until the optimal adsorbent dose and slower increasement of pollutants adsorption rate (almost constant) when linear increasement of adsorbent dose continues after the optimal dose, which authors attributed to the phenomenon of active sites overlaps at higher adsorbent doses [33,35,41–47]. Therefore, based on the results of this part of the study, an MSS dose of 4 g L$^{-1}$ was chosen as the optimal dose for further batch experiments.

### 3.2.2. Effect of Initial N-NO$_3$ Concentrations on Nitrate Removal by MSS

The effect of the initial N-NO$_3$ concentration on the efficiency of nitrate removal by adsorption onto MSS was examined using the initial N-NO$_3$ concentration ranging from 10 to 200 mg L$^{-1}$. From Figure 3, it can be seen that the amount of nitrate uptake strongly depends on the initial nitrate concentration in the solution and increases when increasing the initial nitrate concentration. At the lowest tested initial N-NO$_3$ concentration of 10 mg L$^{-1}$ MSS adsorbed 1.35 mg g$^{-1}$, while at the highest tested initial N-NO$_3$ concentration of 200 mg L$^{-1}$ MSS adsorbed 12.98 mg g$^{-1}$ nitrates from model nitrate solution. Using model wastewater with the same range of the initial nitrate concentrations, a similar uptake was obtained (from 1.00 to 12.16 mg g$^{-1}$). The tests of nitrate removal by MSS from real wastewater samples showed similar MSS efficiencies; the amount of adsorbed nitrate ions increased from 1.02 to 13.70 mg g$^{-1}$ when MSS was tested in wastewater from the confectionary industry, while the nitrate uptake by MSS from meat industry wastewater increased from 0.34 to 12.52 mg g$^{-1}$ when increasing of nitrate concentration from 10 to 200 mg L$^{-1}$.

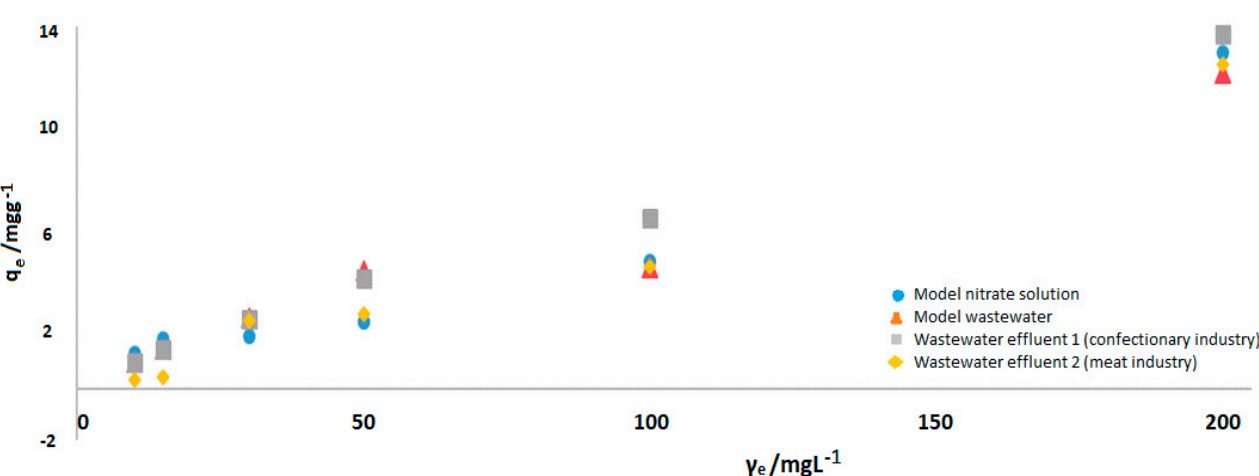

**Figure 3.** Effects of initial N-NO$_3$ concentration on nitrate removal by MSS ($\gamma_{adorbent}$ = 4 g L$^{-1}$, T = 25 °C, V$_{nitrate\ solution}$ = 50 mL, t = 120 min).

This adsorbent behavior is frequent in batch adsorption experiments. The obtained results from this study are coherent with the results of adsorption studies conducted with similar modified materials [33,35,41,44,47,48]. Those studies related the increase of the nitrate ions uptake with the number of available active adsorption sites, increase of the driving force, and concentration gradient at the modified adsorbent surfaces. Keränen et al. [33] used anion exchange-adsorbents obtained by modification of spruce bark, birch bark, pine sawdust, bark, and peat by the ETM method. They reported adsorption capacities for N-NO$_3$ of produced adsorbents in the range from 24.2 to 30.1 mg g$^{-1}$. Stjepanović et al. [35] tested nitrate ions adsorption by brewers' spent grain from various nitrate solutions and reported adsorption capacities ranging from 1.46 to 2.27 mg g$^{-1}$ prior its modification; adsorption capacities ranged from 14.4 to 22.65 mg g$^{-1}$ after chemical modification of brewers' spent grain by the ETM method. Authors of both studies reported increasing the adsorption capacities caused by the increment of the initial nitrate concentration.

3.2.3. Effect of pH on Nitrate Removal by MSS

The pH value of the initial solution can significantly increase or decrease adsorbent efficiency in most of the adsorption processes, and only a smaller number of adsorbents have similar efficiency in a wide pH range [49–51]. Figure 4 shows the effects of initial pH on nitrate removal from four types of water media: nitrate solution, model wastewater, confectionery wastewater, and meat industry wastewater. The adsorption tests were performed in the pH range from 4 to 10. The highest efficiencies of nitrate removal at pH 2 and 4 of 65% and 81%, respectively, were obtained when model wastewater was treated; at all other tested pH, the highest amounts of nitrate were adsorbed by MSS from model nitrate solutions (from 78 to 82%). The lowest percentages of nitrate removal by MSS were obtained at pH 2 in all adsorption tests. Authors of other similar adsorption studies quoted that one of the possible reasons for low uptake at low pH can be caused by the competition of chloride ions for active places at adsorbent surfaces. The authors corroborated those results with the fact that competitive chloride ions were present in all tested water media as the result of pH adjustment with HCl [35,50]. A similar mechanism and the presence of high concentrations of competitive OH$^-$ ions are also denoted as the main reason for low efficiencies when adsorption tests were conducted at a high pH value of 10.

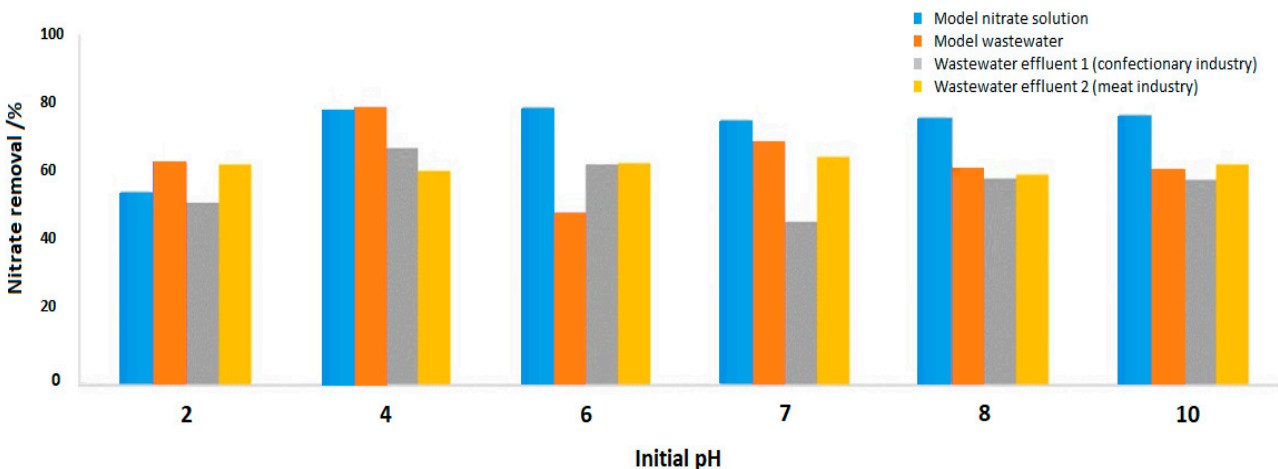

**Figure 4.** Effects of pH on nitrate removal by MSS ($Y_{nitrate}$ = 30 mg L$^{-1}$, $\gamma_{adsorbent}$ = 4 g L$^{-1}$, T = 25 °C, $V_{nitrate\ solution}$ = 50 mL, t = 120 min).

Similar results were reported by Keränen et al. [33]. The authors conducted experiments using pine sawdust and bark, spruce bark, birch bark, and peat and reported that constant nitrate removal rates ranged from 70 to 86% in pH range from 3 to 10 for all five examined materials. However, at pH 2 the nitrate removal efficiencies significantly decreased for all tested materials, except the modified pine sawdust. The authors also reported that the constant and stabilized values of nitrate removal were in the range of 4.4 to 6.0. Xu et al. [39] tested nitrate removal using ETM-modified giant reed and reported an adsorption capacity of 20 mg g$^{-1}$ (about 80% of nitrate reduction) when an initial nitrate solution of 50 mg L$^{-1}$ was tested in the pH range from 4 to 9. The authors emphasized that the ETM modification produces powerful base anion exchangers that work over a wide pH range. Banu and Meenakshi [52] tested the effect of a pH range of 2 to 11 by quaternities melamine formaldehyde resin on nitrate removal and reported that the adsorption capacity for nitrate uptake enlarged from pH 2 to 3 and reached a maximum in the pH range from 3 to 8, while a pH above 8 drastically reduced nitrate adsorption on tested material. The authors explained that at higher pH, the competition between hydroxyl and NO$_3$$^-$ ions for active sites resulted in electrostatic repulsion between the surface sites of the negatively charged adsorbent and the anion, which is not suitable for adsorption processes.

3.2.4. Effect of Contact Time on Nitrate Removal by MSS

The effects of contact time on nitrate removal from the investigated water media were determined in the time range from 2 min to 1440 min using 50 mL of the solution with an initial nitrate concentration of 30 mg L$^{-1}$. The MSS concentration was 4 g L$^{-1}$. Experiments were conducted at 25 °C and the obtained results are presented in Figure 5. From the results presented in Figure 5, it can be observed that the process of nitrate adsorption on MSS can be divided into two phases. The first phase was characterized by fast nitrate adsorption within the initial 30 min of contact between water media and MSS, during which time around 70% of nitrate was removed from model nitrate solution; approximately 60% of nitrates were removed from model wastewater and real wastewater from the confectionary industry and around 30% of nitrate was removed from the meat industry. The second phase was characterized by slower nitrate adsorption until equilibrium was reached (within 60 min).

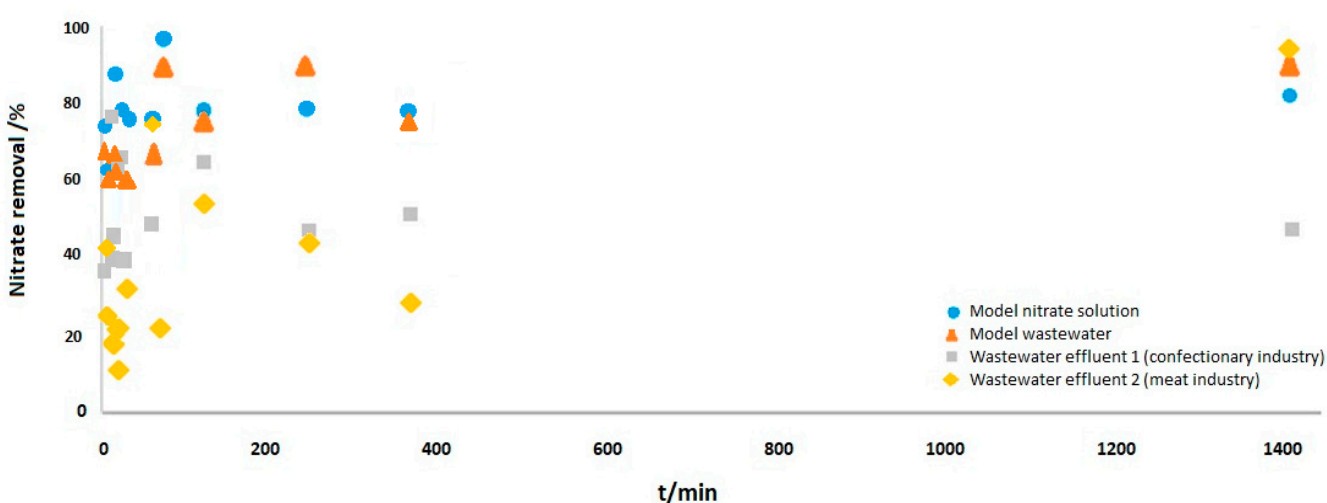

**Figure 5.** Effects of contact time on nitrate removal by MS.

Due to the effect of contact time, the efficiency of nitrate uptake regarding the type of tested water can be defined as follows: model nitrate solution (77.37%) > model wastewater (74.84%) > real wastewater from the confectionary industry (64.24%) > real wastewater from meat industry (53.08%). This order can be explained by the fact that model wastewater and real wastewater samples contain another anion besides nitrates, which competes with nitrate ions for the same adsorption sites. For nitrate adsorption on ETM, MSS ionic exchange was presumably the major responsible mechanism. Our previous study [12] revealed that the adsorption mechanism of nitrate ions onto ETM modification adsorbent is based on the exchange of nitrate ions, present in water, and chloride ions present on the adsorbent surface, which was supported by the results of a similar study [33].

Studies that tested the efficiency of nitrate removal by other types of bioadsorbent reported similar results. Mehdinejadiani et al. [44] reported intensive nitrate uptake by modified wheat straw within the first 10 min of reaction, while Hafshejani et al. [44], based on results obtained in the study of nitrate removal by modified sugarcane bagasse biochar, reported that the presence of carbonate and chloride ions demonstrated the largest and smallest effects on nitrate removal, respectively. The authors also emphasized the fact that in the multi-element solution, the electrostatic interaction of co-existing anions with adsorption sites of modified biochar was much stronger than that of nitrates. They also quoted that a multivalent anion with a higher charge density adsorbs more easily than a monovalent anion. This phenomenon of the ion exchange occurrence prior mechanism of adsorption when ETM-modified biosorbents are used was confirmed by Keränen et al. [33].

### 3.3. Adsorption Equilibrium Modeling

Modeling of experimental adsorption data utilizing convenient isotherm models is essential for the assumption of the adsorption mechanism, which is determined for the practical use of adsorbents in water treatment processes. In this study, Langmuir and Freundlich's models were used to investigate the nitrate adsorption process onto MSS, i.e., distribution of nitrate between liquid and solid phases at equilibrium. Experiments were conducted at three temperature regimes. If data fit more with the Langmuir isotherm model, then the adsorption process can be characterized as monolayer adsorption onto a structurally homogenous adsorbent with energetically equivalent adsorption sites. To fit data by the Langmuir model, Equations (4) and (5) were used. The Freundlich isotherm model implies that the adsorption process occurs on heterogeneous surfaces in the form of the monolayer and/or multilayer formation. For the calculation and data interpretation, Equation (6) was used. Isotherm parameters obtained using the two named models are presented in Table 4.

**Table 4.** Isotherm parameters for nitrate removal by MSS at various temperatures and from various water solutions.

| | | Langmuir | | | | | Freundlich | |
|---|---|---|---|---|---|---|---|---|
| | $T/°C$ | $q_m$ (mg g$^{-1}$) | $K_L$ (L mg$^{-1}$) | $R^2$ | $R_L$ | $n$ | $K_F$ (mg g$^{-1}$)(mg L$^{-1}$)$^{1/n}$ | $R^2$ |
| Model nitrate solution | 25 | 13.351 | 0.017 | 0.3262 | 1.499 | 2.234 | 1.111 | 0.7387 |
| | 35 | 9.737 | 0.030 | 0.4304 | 1.879 | 2.364 | 1.036 | 0.8028 |
| | 45 | 12.063 | 0.028 | 0.4797 | 1.807 | 2.294 | 1.079 | 0.8255 |
| Model wastewater | 25 | 11.779 | 0.029 | 0.6070 | 1.849 | 1.749 | 1.433 | 0.8781 |
| | 35 | 16.611 | 0.032 | 0.9379 | 1.938 | 1.484 | 1.031 | 0.8903 |
| | 45 | 11.261 | 0.046 | 0.7051 | 2.348 | 2.061 | 1.078 | 0.7903 |
| Real wastewater from the confectionary industry | 25 | 18.282 | 0.019 | 0.7539 | 1.744 | 1.504 | 1.623 | 0.9597 |
| | 35 | 31.348 | 0.007 | 0.6499 | 1.274 | 1.184 | 3.505 | 0.9781 |
| | 45 | 14.684 | 0.025 | 0.6527 | 1.979 | 1.877 | 1.160 | 0.9577 |
| Real wastewater from the meat industry | 25 | 7.189 | 0.006 | 0.2680 | 1.231 | 0.754 | 42.052 | 0.8397 |
| | 35 | 7.117 | 0.006 | 0.2299 | 1.231 | 0.753 | 39.591 | 0.8003 |
| | 45 | 12.870 | 0.006 | 0.1625 | 1.231 | 0.771 | 21.546 | 0.8249 |

The dimensionless Langmuir constant, also known as the separation factor ($R_L$), implies the suitability of the isotherm model if $0 < R_L < 1$ [24], while parameter $R^2$ shows the suitability of the isotherm model in the description of adsorption process type. Based on values presented in Table 4, only model wastewater equilibrium data had better compliance with Langmuir isotherm, while all other tested water samples fit better to Freundlich model of adsorption. Calculating from Equation (6), the Freundlich constant $K_F$ at 25 °C had a value of 1.111 when nitrates were adsorbed from model nitrate solution onto MSS, 1.433 when nitrates were adsorbed from model wastewater, and 1.623 when nitrates were adsorbed from real wastewater from the confectionery industry. The highest value of $K_F$ at 25 °C, 42.052, was obtained, then nitrates were adsorbed onto MSS from real wastewater from the meat industry.

Many biosorbents have shown high adsorption capacity for nitrate removal under similar conditions such as amine grafted wheat straw, lauan sawdust, coconut husk, rice husk, pine bark, and sugarcane bagasse [23,39]. Mondal et al. [53] tested nitrate removal using onion peel dust and reported that the equilibrium data were well fitted by the Langmuir adsorption isotherm, which pointed to its maximum adsorption capacity of 5.93 mg g$^{-1}$.

Investigating the applicability of the Freundlich isotherm model, Treybal [54] emphasized that the higher value of n is related to the higher adsorption capacity of the adsorbent. It means that if the value n is between 2 and 10, the adsorption is highly efficient, medium efficient if the value of n is between 1 and 2, and ineffective if the value of n is lower than 1. From Table 4, it can be observed that the values of n are above 2 during the nitrate adsorption from model nitrate solution and above 1 when nitrate was removed from model wastewater and confectionary industry wastewater, which suggested that the nitrate adsorption was effective. Only when the nitrates were adsorbed from real wastewater from the meat industry were the values of n below 1. The dataset shown in Table 4 highlights that the nitrate adsorption on MSS is suitable and that experimental data can be recorded by the Freundlich model.

### 3.4. Adsorption Kinetic Modeling and Mechanism

Adsorption kinetics modeling provides information related to the reaction pathway and mechanism of the process of adsorption [55]. The diffusion process of nitrate uptake on MSS at 25, 35, and 45 °C was analyzed under the following experimental conditions: $\gamma_{nitrate}$ = 30 mg L$^{-1}$, $\gamma_{sorbent}$ = 4 g L$^{-1}$, natural pH. The pseudo-first-order (Equation (7)), pseudo-second-order (Equation (8)), and intraparticle diffusion model (Equation (9)) were used for the adsorption kinetics modeling study of the nitrate adsorption on MSS. Table 5

summarizes the characteristic kinetic parameters and correlation coefficients ($R^2$) obtained from the slope and cross-section of linear surfaces. After comparing the $R^2$ values for used models, it is apparent that the pseudo-second-order kinetic model fits best. $R^2$ values for the pseudo-first-order model are in the rank between 0.0061 and 0.5943, indicating a low correlation.

**Table 5.** Kinetics parameters of nitrate adsorption on MSS.

| T/°C | | Pseudo-First Order | | | Pseudo-Second Order | | |
|---|---|---|---|---|---|---|---|
| | | $q_{e\,cal}$ (mg g$^{-1}$) | $k_L$ (min$^{-1}$) | $R^2$ | $q_{e\,cal}$ (mg g$^{-1}$) | $k_L$ (g mg$^{-1}$)(min$^{-1}$) | $R^2$ |
| Model nitrate solution | 25 | 0.998 | 0.716 | 0.3337 | 2.851 | 0.569 | 0.999 |
| | 35 | 0.999 | 0.380 | 0.0923 | 2.721 | 1.716 | 0.999 |
| | 45 | 0.996 | 0.979 | 0.5836 | 2.758 | 0.389 | 0.999 |
| Model wastewater | 25 | 0.996 | 0.014 | 0.3390 | 2.929 | 0.057 | 0.987 |
| | 35 | 0.999 | 0.265 | 0.1256 | 2.630 | 0.471 | 0.992 |
| | 45 | 1.019 | 1.034 | 0.2373 | 2.765 | 0.165 | 0.995 |
| Real wastewater from the confectionary industry | 25 | 0.999 | 0.282 | 0.0545 | 1.714 | 0.329 | 0.981 |
| | 35 | 1.001 | 0.139 | 0.0493 | 1.570 | 0.120 | 0.978 |
| | 45 | 1.002 | 0.038 | 0.0884 | 1.316 | 0.070 | 0.970 |
| Real wastewater from the meat industry | 25 | 0.999 | 1.037 | 0.0493 | 1.639 | 0.186 | 0.860 |
| | 35 | 0.999 | 1.071 | 0.0061 | 1.177 | 0.328 | 0.945 |
| | 45 | 0.988 | 0.572 | 0.5943 | 1.093 | 0.204 | 0.890 |

Pseudo-second-rate constants were calculated using Equation (10). From the data in Table 5, it can be observed that the q$_{ecal}$ values counted using the pseudo-second-order model were very near to the experimental q$_{e\,exp}$ values. Results also show that the pseudo-second-order model better describes the experimental kinetics data for nitrate removal, which correlates with the results of similar studies conducted to examine nitrate removal using biosorbents [49,51,56].

Table 6 shows that the values for k$_{l1}$ were 0.104 mg g$^{-1}$ min$^{-0.5}$ in model nitrate solution, 0.038 mg g$^{-1}$ min$^{-0.5}$ in model wastewater, 0.047 mg g$^{-1}$ min$^{-0.5}$ in real wastewater from the confectionery industry, and 0.083 mg g$^{-1}$ min$^{-0.5}$ in real wastewater from the meat industry at 25 °C. The second phase demonstrates equilibrium due to low concentrations of adsorbates in the solution, which could elucidate the lower k$_{l2}$ from 0.004 to 0.084 mg g$^{-1}$ min$^{-0.5}$.

**Table 6.** Intraparticle diffusion model parameters of nitrate adsorption on MSS at various temperatures.

| T/°C | | $k_{i1}$ (mg g$^{-1}$min$^{0.5}$) | $c_1$ (mg g$^{-1}$) | $R_1^2$ | $k_{i2}$ (mg g$^{-1}$min$^{0.5}$) | $c_2$ (mg g$^{-1}$) | $R_2^2$ |
|---|---|---|---|---|---|---|---|
| Model nitrate solution | 25 | 0.104 | 2.361 | 0.3005 | 0.009 | 2.702 | 0.7737 |
| | 35 | 0.117 | 2.196 | 0.6484 | 0.004 | 2.655 | 0.0253 |
| | 45 | 0.106 | 2.135 | 0.8639 | 0.007 | 2.702 | 0.0721 |
| Model wastewater | 25 | 0.038 | 2.229 | 0.5151 | 0.061 | 1.777 | 0.7312 |
| | 35 | 0.002 | 2.642 | 0.0014 | 0.004 | 2.555 | 0.0189 |
| | 45 | 0.009 | 2.587 | 0.0291 | 0.054 | 1.681 | 0.7399 |
| Real wastewater from the confectionary industry | 25 | 0.047 | 1.475 | 0.1964 | 0.039 | 2.378 | 0.2901 |
| | 35 | 0.040 | 1.961 | 0.0007 | 0.046 | 2.374 | 0.3259 |
| | 45 | 0.006 | 1.897 | 0.0009 | 0.084 | 2.799 | 0.5957 |
| Real wastewater from the meat industry | 25 | 0.083 | 1.052 | 0.5431 | 0.064 | 3.089 | 0.0895 |
| | 35 | 0.082 | 0.709 | 0.4763 | 0.073 | 2.612 | 0.0641 |
| | 45 | 0.006 | 1.135 | 0.0100 | 0.026 | 1.843 | 0.0235 |

### 3.5. Breakthrough and Desorption Studied

A fixed-bed column was used to examine the MSS efficiency for nitrate removal under static conditions. The experiments were conducted with a continuous flow rate of 10 mL min$^{-1}$ and a column loading with 1 g of adsorbent and obtained data as the breakthrough curves are presented in Figure 6. One of the most important characteristics of a promising adsorbent is good adsorption characteristics and efficient re-usage after multiple adsorption-desorption cycles and regeneration [9,56]. Obtained results are presented in Table 7 and are compared with the maximum adsorption capacities of MSS for nitrate removal obtained by dynamic conditions (batch experiments). From Table 7, it can be observed that nitrate removal using MSS in fixed-column obtained higher values; this can be related to high-density packing form and slow leaking of water media through the column.

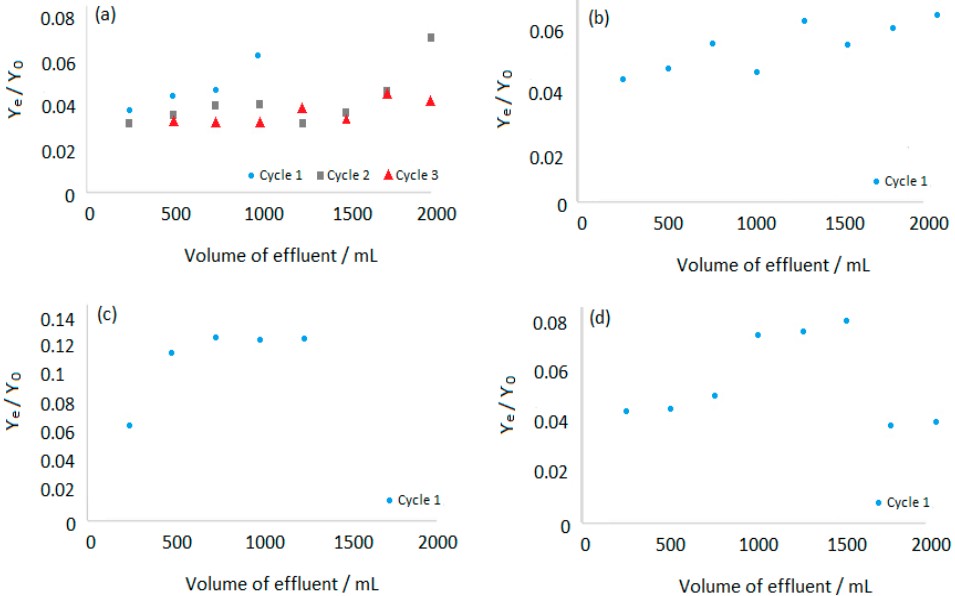

**Figure 6.** Breakthrough curves for the adsorption of nitrate ions from (**a**) model nitrate solution, (**b**) model wastewater, (**c**) real wastewater from the confectionery industry, and (**d**) real wastewater from the meat industry on MSS.

**Table 7.** MSS adsorption capacities under dynamic (batch experiments) and static (fixed-bed column) conditions.

| | Adsorption Capacitiy (mg g$^{-1}$) | |
|---|---|---|
| | Batch experiments | Fixed-bed column |
| Model nitrate solution | 13.351 | 43.38 |
| Model wastewater | 9.737 | 24.59 |
| Real wastewater from the confectionary industry | 12.063 | - |
| Real wastewater from the meat industry | 11.779 | 33 |

In this study, three adsorption and desorption cycles were carried out to analyze the possibility of MSS reuse. The adsorption phases were carried out using the model nitrate solution; after the column was completely saturated, the regeneration was carried out using 0.1 M NaCl solution at a flow rate of 10 mL min$^{-1}$. Then, after correctly rinsing the column with demineralized water, a second cycle was carried out. After the second and third cycles (Figure 6a), regeneration did not adversely affect the adsorption sites, i.e., the studied material remained stable and reusable. The saturation absorption capacity after the first cycle for MSS was 37.53 mg g$^{-1}$, while after the second and the third cycle it was

43.38 mg g$^{-1}$ and 40.95 mg g$^{-1}$, respectively. This demonstrates that the MSS still removes nitrate ions effectively after the regeneration treatment. The desorption experiments were also conducted using the other tested types of wastewater (Figure 6b–d).

However, obtained results showed significantly lower adsorption capacities of MSS when all types of tested wastewater were used and only one adsorption cycle was conducted before the columns were saturated. Testing the adsorption capacity of MSS using model wastewater up to 24.59 mg g$^{-1}$ was adsorbed. The column test using the confectionery industry wastewater was not carried out efficiently.

Due to the complex compositions of used real wastewater (fat molecules and other insoluble pollutants), after 1250 mL of water feed, the column was clogged. The data for the experiment of desorption utilizing the real wastewater from the meat industry showed 33 mg g$^{-1}$ saturation capacities, but after the regeneration procedure column was quickly clogged (Figure 6d). Therefore, a wastewater pretreatment, like filtration, could efficiently contribute to column clogging avoidance.

Therefore, a wastewater pretreatment, like filtration, could efficiently contribute to column clogging avoidance.

### 3.6. Determination of Acute Toxicity of MSS Samples by Daphnia Magna Toxicity Tests

After adsorption treatment, samples of used MSS were ecotoxicologically tested to determine the acute toxicity. Each set of ecotoxicological tests was conducted using 20 model organisms of the freshwater plankton *Daphnia magna*. The test included the determination of plankton immobilization after exposure to the solutions prepared by various amounts of MSS used for wastewater treatment. The acute toxicity of saturated MSS was determined after 24 or 48 h of organism exposure using the method HRN EN ISO 6341:2013. The immobilization was considered as plankton's inability to move even after a slight mixing of the test solutions. The obtained results are shown in Tables 7 and 8.

From Table 8, it can be observed that after 24 h of organisms' exposure to the 2% MSS solution, the highest nitrate impact (25% of immobilized *Daphnia magna*) was observed when MSS was saturated with nitrates from the nitrate model solution. The same effect of the highest number of immobilized organisms was also obtained after 48-h tests when 45% of organisms were immobilized. Acute toxicity tests conducted with MSS treated with real wastewater showed lower percentages of immobilized *Daphnia*.

**Table 8.** The acute toxicity of nitrate-saturated MSS was determined by *Daphnia magna* toxicity tests.

| Nitrate Saturated MSS | Solution | Immobilization | |
|---|---|---|---|
| | % | 24 h | 48 h |
| MS | 0.5 | 20 | 30 |
| | 1 | 25 | 35 |
| | 2 | 25 | 45 |
| CW | 0.5 | 0 | 20 |
| | 1 | 10 | 30 |
| | 2 | 20 | 40 |
| MW | 0.5 | 5 | 10 |
| | 1 | 10 | 20 |
| | 2 | 20 | 40 |

No effects of *Daphnia* immobilization after 24 h were observed when a test was conducted with 0.5% solution of MSS treated by real wastewater from the confectionery industry used, while immobilization after 48 h was lowest (10%) when 0.5% solution of MSS treated by real wastewater from meat industry was used. From the results of acute toxicity of used MSS samples, it can be concluded that discharging of used MSS adsorbents would have the lowest negative environmental impact if MSS were used for nitrate removal from real wastewater samples than MSS used for nitrate removal from nitrate model solutions. This conclusion is also based on the results presented in the previous section,

where breakthrough and desorption tests showed the highest mobility of nitrates adsorbed onto MSS from nitrate model solutions.

## 4. Conclusions

This study examined nitrate removal from wastewater using the ETM-chemically modified sunflower seed shells, a low-cost lignocellulose material obtained as food industry byproduct. The MSS performances regarding nitrate uptake were tested under dynamic (batch test) and static conditions (fixed-bed column).

Raw (SS) and modified sunflower seed shells (MSS), prior and after nitrate adsorption, were chemically and structurally analyzed by energy-dispersive X-ray spectroscopy (EDS), Field Emission Scanning Electron Microscope (SEM) and Transmission Electron Microscope (TEM). The FESEM analyses reviled significant differences in morphology and structure among SS and MSS.

Nitrate adsorption efficiency of MSS was tested using four water media: model water and wastewater and two real wastewater (confectionery and meat industry). Nitrate adsorption by MSS achieved equilibrium within 60 min at all tested conditions, while the highest efficiency in nitrate removal was achieved at pH 7 for the wastewater from meat industry, while pH 4 was the best for model wastewater, and pH 6 for a model nitrate solution.

Examining the influence of temperature on the adsorption process, it was determined that the highest adsorption capacities were achieved at 35°C. Regarding adsorption efficiency, the best results were achieved with model wastewater 86.37%, 86.36% with confectionery industry wastewater, 86.12% with nitrate model solution and 66.00% with meat industry wastewater. The Freundlich model of isotherm adsorption showed a preferable match to the adsorption data, suggesting the process of multilayer adsorption.

The kinetic study of nitrate adsorption on MSS show that pseudo-second-order kinetic models describe it. The processing of kinetic data by the kinetic model of intraparticle diffusion shows that adsorption takes place through two steps, that is, intraparticle diffusion is not the only process that controls nitrate adsorption on modified adsorbents.

The column tests showed that MSS could be successfully regenerated only when nitrates were adsorbed from a model nitrate solution. However, when MSS was used for nitrate removal from model wastewater and real wastewaters (confectionary and meat industry) column clogging was observed.

Ecotoxicological acute tests conducted to determine the acute toxicity of nitrate saturated MSS to the freshwater plankton *Daphnia magna* showed that discharging of used MSS would have minimal negative environmental impact if MSS were used for treatments of real wastewater samples.

**Author Contributions:** Conceptualization, M.H-S.; methodology, M.H-S., H.D., M.E.R., A.T., Ž.R. and M.Š.; software, H.D., T.L.D. and M.Š.; validation, A.K.J., M.H.-S. and A.T.; formal analysis, A.K.J., H.D., A.T. and Ž.R.; investigation, A.K.J., M.H.-S. and H.D.; resources, A.K.J., M.H.-S., H.D. and Ž.R.; data curation, M.H.-S., H.D., M.E.R. and M.Š.; writing—original draft preparation, A.K.J.; writing—review and editing, M.H.-S. and H.D.; visualization, A.K.J., H.D. and M.H.-S.; supervision, M.H.-S. All authors have read and agreed to the published version of the manuscript.

**Funding:** This research received no external funding.

**Data Availability Statement:** The data presented in this study are available upon request from the corresponding author.

**Conflicts of Interest:** The authors declare no conflict of interest.

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
