# Peer review of "Utilization of Modified Sunflower Seed as Novel Adsorbent for Nitrates Removal from Wastewater"

_water, doi:10.3390/w16010073_

Round 1

Reviewer 1 Report

Comments and Suggestions for Authors

The article discusses the synthesis of modified sunflower seed for application in nitrate removal from wastewater. The modified sunflower seed showed vast improvements in terms of its structural as well as its performance for the said application, compared to the unmodified sunflower seed. Overall, the content of the paper is sufficient, showing good flow in expressing the core idea of the study carried out. For the most part the figures and table are clear and concise. There are several suggestions that we would like to propose prior to the paper being published:

1.     The proposed title is confusing. It is as if the sunflower seed has been modified with an agriculture waste, while the agriculture waste is referring to the sunflower seed.

2.     Consistency issue – the terms “adsorber” or “adsorbent”

3.     Irrelevant and inconsistent uses of abbreviations – please check.

4.     Line 44 – “.., and similar.” What do you mean by similar? Revise.

5.     Line 64-65 – “… due to their .. ecological impact.” Revise the sentence.

6.     In the paragraph from line 71 to 79, there are three issues: 1) is activated carbon expensive? 2) Is the process for the separation of absorbent from solution being addressed in this study? 3) What is the targeted value of adsorption capacity of adsorbent towards nitrate?

7.     Model wastewater was made by Kosjek et al. [13]. What do you mean by this?

8.     Section 2.4 – how did you separate the adsorbent from the nitrate solution after the adsorption process for the determination of nitrate concentration?

9.     Line 289-291 – I am not sure what the authors are trying to mention here.

10.  Figure 3, 4, 6 – Why there are error bars for the x-axis? Were the manipulated variables fluctuating?

11.  Overall, the discussions are clear. However, we would like to suggest for the authors to discuss in further detail how the improvement in the properties of modified sunflower seed is related to its improved performance. At present, the explanation could be further elaborated especially in terms of the overall mechanism of the process taking place during the experiment.

12.  It would be better if the authors compare the nitrate adsorption capacity of the modified sunflower seed with the materials developed in other studies.

Comments on the Quality of English Language

Satisfactory.

Author Response

Dear Reviewer 1,

authors of the manuscript water-2725663 are very thankful for your comments, suggestions, and additional questions.

Here are our answers:

  1. The proposed title is confusing. It is as if the sunflower seed has been modified with an agriculture waste, while the agriculture waste is referring to the sunflower seed.

The title of the manuscript is revised.

  1. Consistency issue – the terms “adsorber” or “adsorbent”

The term is unified.

  1. Irrelevant and inconsistent uses of abbreviations – please check.

The check is done and corrections are made.

  1. Line 44 – “.., and similar.” What do you mean by similar? Revise.

The revision has been made.

  1. Line 64-65 – “… due to their .. ecological impact.” Revise the sentence.

The sentence is revised.

  1. In the paragraph from line 71 to 79, there are three issues: 1) is activated carbon expensive? 2) Is the process for the separation of absorbent from solution being addressed in this study? 3) What is the targeted value of adsorption capacity of adsorbent towards nitrate?

The text is corrected. The data about the adsorption capacities of similarly developed adsorbents is given in Table 1.

  1. Model wastewater was made by Kosjek et al. [13]. What do you mean by this?

The sentence is corrected. 

  1. Section 2.4 – how did you separate the adsorbent from the nitrate solution after the adsorption process for the determination of nitrate concentration?

The text in Section 2.4. has been revised.

  1. Line 289-291 – I am not sure what the authors are trying to mention here.

The text between lines 289-291 has been revised.

  1. Figure 3, 4, 6 – Why there are error bars for the x-axis? Were the manipulated variables fluctuating?

Figures 3,4 and 6 are corrected according to the obtained results (there no manipulated variables fluctuating).

  1. Overall, the discussions are clear. However, we would like to suggest for the authors to discuss in further detail how the improvement in the properties of modified sunflower seed is related to its improved performance. At present, the explanation could be further elaborated especially in terms of the overall mechanism of the process taking place during the experiment.

An additional explanation of the adsorption mechanism is given in Section 3.2.4.

  1. It would be better if the authors compare the nitrate adsorption capacity of the modified sunflower seed with the materials developed in other studies.

The list of adsorption capacities of several materials developed in other studies for the comparison of the obtained adsorption capacities is given in Table 1.

Reviewer 2 Report

Comments and Suggestions for Authors

The presented manuscript includes the study of the utilization of agriculture waste - modified sunflower seed as novel adsorber for nitrates removal from wastewater.

The paper is of interest. The results of the work are presented on a good level but some questions and weak points should be mentioned.

1. Title. Change “adsorber” to “adsorbent”.

2. Introduction. Authors should give more discussion and examples about the materials for the sorption of nitrates. 

3. Fig.1. Instead of unuseful EDS spectra (regular for low-impact papers because EDS spectra are an intermediate result), the best way is to show results in a table with standard deviation ranges.

4. Section 2.3. Looks absolutely frustrating analysis of Nb, Rh, Zr, Cu, Rb in obtained samples. Please do not scare readers and reviewers, and delete these elements.

5. Porous materials need to have a BET analysis and pore size distribution, as high surface area and suitable pore sizes are important for sorption studies.

6. Fig. 3, 4, 6. Why do authors show SD ranges for the X-axis? Didn’t they know the exact dosage of adsorbent used in the experiments?

7. How many experiments were done in parallel? Please add standard deviations everywhere where applicable.

8. Most of the references are too old. Please, up to date the relevant references for the 2023 and 2024 years to prove the novelty and actuality of the work.

9.  Obtained results have to be compared with published analogs through the results and discussion parts.

Author Response

Dear Reviewer 2,

authors of the manuscript water-2725663 are very thankful for your comments, suggestions, and additional questions.

Here are our answers:

  1. Change “adsorber” to “adsorbent”.

The change has been made.

  1. Introduction. Authors should give more discussion and examples about the materials for the sorption of nitrates.

Additional text and Table 1 are given in the Section 1. Introduction.

  1. Fig.1. Instead of unuseful EDS spectra (regular for low-impact papers because EDS spectra are an intermediate result), the best way is to show results in a table with standard deviation ranges.

EDS spectra are removed. The results of the EDS analysis along with additional data are presents in Table 3.

  1. Section 2.3. Looks absolutely frustrating analysis of Nb, Rh, Zr, Cu, Rb in obtained samples. Please do not scare readers and reviewers, and delete these elements.

The text in Section 2.3 is revised.

  1. Porous materials need to have a BET analysis and pore size distribution, as high surface area and suitable pore sizes are important for sorption studies.

Unfortunately, we are not currently able to perform BET analysis. Thank you for understanding.

  1. Fig. 3, 4, 6. Why do authors show SD ranges for the X-axis? Didn’t they know the exact dosage of adsorbent used in the experiments?

Figures 3,4 and 6 are corrected according to the obtained results (the exact dosage of the adsorbent used in the experiment was known. Misinterpretation of the obtained results).

  1. How many experiments were done in parallel? Please add standard deviations everywhere where applicable.

All tests were performed in duplicate and found to be repeatable.

  1. Most of the references are too old. Please, up to date the relevant references for the 2023 and 2024 years to prove the novelty and actuality of the work.

A novel references related to our study are used to improve this manuscript.

  1. Obtained results have to be compared with published analogs through the results and discussion parts.

The obtained results are additionally compared with the results of similar studies.

Reviewer 3 Report

Comments and Suggestions for Authors

This manuscript proposed utilization of agriculture waste-modified sunflower seed as novel adsorber for nitrates removal from wastewater. A systematic experimental study was carried out. We will recommend its acceptance for the journal “Water” if the authors solve the following problems.

1. Line 94, What is SS? Should the abbreviation be explained in brackets for the first time?

2. Line 153, the word “temp, and era”—is the word wrong.

3. The adsorption mechanism should be further explained, and it is better to draw a mechanism diagram to more intuitively explain the adsorption of nitrates by modified sunflower seed. 

4. In this paper, what is the optimal adsorption condition (pH, contact time, temperature) for the treatment of nitrates in model wastewater, confectionery industry, wastewater, and meat industry?

5. In the actual industry, how to control the production cost for the application of the modified sunflower seed?

6. What is the adsorption efficiency of nitrates in the model wastewater, confectionery industry, wastewater, and meat industry under static and dynamic conditions, respectively?

7.  Figure 1, Figure 3, Figure 4, Figure 5, Figure 6, and Figure 7 are blurred and not clear. It is recommended to change the picture format or redraw.

Author Response

Dear Reviewer 3,

authors of the manuscript water-2725663 are very thankful for your comments, suggestions, and additional questions.

  1. Line 94, What is SS? Should the abbreviation be explained in brackets for the first time?

The change of the text has been made.

  1. Line 153, the word “temp, and era”—is the word wrong.

The sentence is changed.

  1. The adsorption mechanism should be further explained, and it is better to draw a mechanism diagram to more intuitively explain the adsorption of nitrates by modified sunflower seed. 

Additional explanation of the adsorption mechanism is given in Section 3.2.4.

  1. In this paper, what is the optimal adsorption condition (pH, contact time, temperature) for the treatment of nitrates in model wastewater, confectionery industry, wastewater, and meat industry?

Additional text about optimal adsorption conditions is added in the section Conclusion (line 607-613).

  1. In the actual industry, how to control the production cost for the application of the modified sunflower seed?

Additional text about adsorbent costs is added in the section Introduction (line 93-105).

  1. What is the adsorption efficiency of nitrates in the model wastewater, confectionery industry, wastewater, and meat industry under static and dynamic conditions, respectively?

A comparison of obtained values under static and dynamic conditions is given in Section 3.5.

  1. Figure 1, Figure 3, Figure 4, Figure 5, Figure 6, and Figure 7 are blurred and not clear. It is recommended to change the picture format or redraw.

The quality of all listed Figures is improved.

Round 2

Reviewer 2 Report

Comments and Suggestions for Authors

all comments were addressed